# Surface-Enhanced Raman Scattering Detection of Fipronil Pesticide Adsorbed on Silver Nanoparticles

**DOI:** 10.3390/s19061355

**Published:** 2019-03-18

**Authors:** Nguyễn Hoàng Ly, Thi Ha Nguyen, Ngô Đình Nghi, Young-Han Kim, Sang-Woo Joo

**Affiliations:** 1Department of Chemistry, Soongsil University, Seoul 156-743, Korea; nguyenhoangly2007@gmail.com (N.H.L.); hakhtn94@gmail.com (T.H.N.); nghingo1993@gmail.com (N.Đ.N.); 2Department of Information Communication, Materials, Chemistry Convergence Technology, Soongsil University, Seoul 156-743, Korea; younghak@ssu.ac.kr

**Keywords:** fipronil, Raman spectroscopy, DFT calculations, food contaminations

## Abstract

This work presents a surface-enhanced Raman scattering (SERS) and density functional theory (DFT) study of a fipronil adsorbed on colloidal silver nanoparticles (AgNPs). A standard curve was established to quantify fipronil within a range of 0.0001–0.1 ppm (r^2^ ≥ 0.985), relying on the unique fipronil Raman shift at ~2236 cm^−1^ adsorbed on AgNPs. DFT calculations suggest that the nitrile moiety (C≡N) binding should be slightly more favorable, by 1.92 kcal/mol, than those of the nitrogen atom of the pyrazole in fipronil and Ag_6_ atom clusters. The characteristic peaks of the SERS spectrum were identified, and both the calculated vibrational wavenumbers and the Raman intensity pattern were considered. The vibrational spectra of fipronil were obtained from the potential energy distribution (PED) analysis and selective Raman band enhancement.

## 1. Introduction

Fipronil, (5-amino-1-[2,6-dichloro-4-(trifluoromethyl) phenyl]-4-[(trifluoromethyl)-sulfinyl]-1*H*-pyrazole-3-carbonitrile), is a potent insecticide of the phenylpyrazole group that is widely used in agriculture to control pests with high efficiency at very low concentrations [1]. Fipronil can impact γ-aminobutyric acid reception in nerve transmission and effectively block γ-aminobutyric acid–regulated chloride channels in the nervous system, paralyzing or killing the target organism [2]. Fipronil in food mixtures is also known to cause serious diseases in the human body. Fipronil exposure changes blood biochemistry and sex hormone levels; it can also lead to reductions in the cellular immune response, antioxidant abilities, and carotenoid-reliant coloration [3]. 

Endocrine disruption occurs due to alterations in gene expression in *Callinectes sapidus* populations exposed to environmental levels of fipronil [4]. In 2017, according to a report by the European Commission’s Health and Food Safety Directorate-General, levels of fipronil had been detected as high as 0.72 mg/kg in eggs and 0.77 mg/kg in chicken meat [5]. There are therefore many researchers interested in the residue analysis of fipronil in chicken eggs and muscles by liquid chromatography–tandem mass spectrometry (LC-MS/MS) [2,6,7], or fipronil detection using immunoassay [8,9] and gas chromatography [10]. Although the detection limit is extremely low, sample preparation is complex and time-consuming.

Recently, the scientist’s community has paid much attention to the spectroscopic detection of hazardous pesticide [11,12,13]. Raman methods have the advantage of easy detection in aqueous solutions with tremendous electromagnetic field enhancements on metal surfaces to provide extremely high sensitivity to monitor organic contaminants [14,15]. Density functional theory (DFT) can be introduced to interpret Raman spectral features to understand the quantum mechanical properties of small organic molecules adsorbed on metal atoms [16].

Surface-enhanced Raman scattering (SERS) procedures for the detection of fipronil are unavailable in the literature, despite SERS being conventionally applied to biosensors [17,18,19], and a possible way of detecting trace amounts of toxic contaminants in minimally processed food products [20]. Due to significant enhancements and resulting sensitivity, SERS has advantages over conventional methods (e.g., facile sample preparations and on-site detection). Because of irregular hot spots and equivocal selection rules, it is essential to identify novel nanostructures, such as oxide thicknesses on a Si/SiO_2_ [17] substrate or silver nanoparticles (AgNPs) [19,21,22], and convenient methods to increase sensitivity. The ability to increase both the sensitivity and selectivity of the SERS analytical method is the main reason for its role as a primary emergent spectral technique that has found many applications with food, such as monitoring and quality control in industrial food manufacture, food safety in agricultural plant production [23], and detection of pesticide residues in food [24]. 

DFT calculations have also seen significant advances in a wide range of molecular properties [25], allowing a close connection between theoretical and experimental research and often producing important evidence about the geometric shape, electronic detail, and spectroscopic properties of the issues being studied [26,27]. Therefore, the development and application of systems based on combinations of SERS and DFT calculations are a current hot topic that has contributed to a better understanding of the phenomena related to detection and monitoring at very low concentrations—especially at physiological levels [18,22,24]. In particular, a great deal of effort has been reported toward developing a simple and rapid method for detecting the pesticide fipronil on eggshells and in liquid eggs by Raman spectroscopy [5]. However, some limitations associated with previously reported sensors remain, and more facile assay techniques for fipronil detection are needed to overcome them. Notably, there has been no recent significant consideration of SERS application combined with DFT calculations for fipronil detection.

The maximum fipronil level allowed in food (rice and corns) by the United States Food and Drug Administration and the European Commission is 0.01 ppm. Despite previous report [5] on the detection of fipronil using Raman spectroscopy or other methods, a more efficient nanostructure platform, in terms of facile usage and improved sensitivity, would still be helpful in analytical food and agriculture science. In this study, we report the detection of fipronil using SERS on AgNPs, considering both the calculated vibrational wavenumbers and Raman intensity patterns. Our method can usefully detect fipronil in food, such as eggshells and liquid eggs, and in the water and soil environment.

## 2. Materials and Methods

### 2.1. Materials 

Silver nitrate (99%), trisodium citrate (>99%), and cetyltrimethylammonium chloride (CTAC) solution (25 wt% in H_2_O) were purchased from Sigma-Aldrich (St. Louis, MI, USA). Fipronil pesticide was purchased from Tokyo Chemical Industry (Tokyo, Japan).

### 2.2. Preparation of AgNPs

The synthesis of AgNPs was performed according to previous work [28], with a slight modification. We prepared the AgNPs using a citrate reduction method; a predetermined amount of triple distilled water (50 mL) and a solution of precursors (1 mL AgNO_3_ 55 mM) was intermixed, disturbed, and heated to the boiling point of water. A suitable amount of sodium citrate solution (1 mL, 0.57 g/50 mL) was then promptly admixed to this mixture and continuously disturbed via magnetic stir bar while the mixture was boiling. Finally, small amounts of the triple-distilled water were admixed to the reaction solution slowly and continuously, so as to maintain the initial level of the solution, for one hour. According to measurements by a zeta-potential analyzer (ELS-Z, Otsuka Electronics, Osaka, Japan), and a high-resolution transmission electron microscope (JEM-3100, JEOL, Tokyo, Japan), the diameter of the AgNPs obtained from this procedure was approximately 50 nm. 

### 2.3. DFT Calculations

DFT calculations were performed similarly to those we recently reported [29], and potential energy distribution (PED) calculations were conducted based on previous literature [30]. The quantum chemical density functional calculations were performed using Gaussian 09 analytical suite software [31]. The computation of Raman vibrational frequencies was performed following the energy optimization of the fipronil and was determined by conducting DFT calculations, at the RB3LYP/6-31G(d,p) level of theory on the free molecule. Subsequently, AgNP modeling was conducted using the RB3LYP/LANL2DZ method to optimize Ag_3_^+^, Ag_4_^+^, and Ag_6_ cluster geometry prior to binding with the fipronil molecule, then the whole complex of the silver-fipronil was fully optimized again. Finally, the Raman vibrational assignments of fipronil were obtained based on PED calculations and analyzed in detail with the vibrational energy distribution analysis (VEDA) program, which directly and automatically read the input data from the Gaussian program output files.

### 2.4. Instrumentations

UV-Vis absorption spectral changes of the fipronil, before and after applying the AgNP colloidal solution, were obtained with a spectrophotometer (Optizen 3220UV, Mecasys, Daejeon, Korea). All Raman data were obtained using a Raman microscope system (RM 1000, Renishaw, Wotton-under-Edge, UK) with either a 633 nm HeNe excitation laser or a 514 nm multiline Argon ion laser (25-LHP-928, 35-LAP-431-220, Melles Griot, Albuquerque, NM, USA) and a charge-coupled device (CCD) camera. The SERS spectra were recorded via spectroscopic glass tubes after the preparation of the fipronil-Ag samples with a x20 objective lens. The integration time for the SERS measurement was set at 10 s per spectrum with a range of 200–3200 cm^−1^. Prior to performing SERS, the spectral positions were calibrated based on the Si peak at 520 cm^−1^.

### 2.5. SERS Preparation of Fipronil on AgNPs 

For the SERS experiment on fipronil adsorbed on colloidal AgNPs, 1000 µL of the liquid bearing these synthesized AgNPs was first put into a 1.5 mL centrifugal tube and then centrifuged at 10,000 rpm at 4 °C for 10 min. Subsequently, 920 µL of the supernatant was carefully removed and the remaining 80 µL of the AgNPs aqueous solution was mixed with 10 µL of fipronil solution (2000 ppm [*m*/*v*] in ethanol). Due to the poor solubility of fipronil in aqueous solutions, 10 µL of CTAC was also added to improve dispersibility. Finally, the 100 µL mixture (containing about 200 ppm of fipronil) was sonicated for 30 s and kept stable for 15 min at room temperature, and the SERS spectra of the 100 µL of the AgNPs-fipronil complex were recorded. 

## 3. Results and Discussion

### 3.1. Normal Raman Spectra of Fipronil

One of the main advantages of Raman spectroscopy is providing detailed structural information, making it a powerful tool for molecular vibration investigation of the analyte being studied without the need for additional labeling. However, only DFT quantum chemical calculations of structural features and selected physicochemical properties of fipronil have previously been reported, notably those for solid fipronil [25]. Therefore, for the purposes of fipronil detection using the SERS spectra, we present the simulation and experimental results of the Raman spectrum for solid fipronil in Figure 1b. As shown in Figure 1b, the most intense band of fipronil is found at 2248 cm^−1^ for the experimental measurement and at 2301 cm^−1^ for the calculation, which is the symmetric stretching vibration of the C≡N group. The characteristic bands of the fipronil molecule are also listed in Table 1.

The optimized structural representation of this analyte in its neutral form is shown in Figure 1a, with observed discrepancies of 53 cm^−1^, on average, between the experimentally determined and estimated Raman vibrational values in Figure 1b, thus demonstrating a relatively good agreement between these two outcomes, with the exception of some variations in the peak intensities. However, scaling factors have usually been used in the literature to account for systematic empirical errors having specified initial values from the force field constants in the employment of quantum mechanical approaches. These were not applied in this case to the scaling factor of the simulated frequencies to improve peak position agreements, specifically by reducing the discrepancies at higher frequencies.

### 3.2. SERS Spectra of Fipronil on AgNPs

SERS signals of fipronil adsorbed on the surface of the AgNPs could not be obtained, even when screening high concentrations (100–1000 ppm) of pure fipronil [5]. This was mainly due to the weak interaction between fipronil and the AgNPs, and the low solubility of fipronil, which can easily crystallize out of aqueous solutions. To overcome these limitations, a cationic surfactant, CTAC, was added to the mixture solution of fipronil and AgNPs to improve the dispersibility of fipronil in aqueous solutions. Figure 2a illustrates a plausible mechanism for the adsorption of fipronil molecules on AgNP surfaces using CTAC linking. The surfaces of plasmonic AgNPs are modified by the CTAC molecules through electrostatic interactions between the positively charged CTAC and the negatively charged AgNPs. CTAC is a cationic surfactant that may be usable in the modification of the surface charge of AgNPs; it could also be used as a dispersant to prevent the aggregation of AgNPs, while increasing the solubility of fipronil in solution. The binding between fipronil and AgNPs in the presence of CTAC is easier than in the absence of CTAC, due to electrostatic interactions between the positively charged CTAC-modified AgNPs and the negatively charged fipronil. In general, fipronil molecules interacting with the surface of AgNPs through nitrogen atoms do so more strongly than other atoms in fipronil molecules. The atomic charges of these nitrogen atoms, such as N_19_, N_20_, and N_24_ (atomic numbering depicted in Figure 1a), were previously estimated at −0.58, −0.77, and −0.56 (a.u.), respectively, after fully unconstrained geometry optimization in aqueous phases [32]. Our DFT calculations of the atomic charges of these nitrogen atoms were also consistent with those data. Thus, we can predict that the charge of fipronil is negative in the interaction with the AgNPs surface. Moreover, CTAC has a chloride ion in molecular which could potentially lead to high SERS enhancements. The chloride ion is well known for its activation effect on the SERS spectra of aromatic molecules, such as phthalazine and pyridine, adsorbed on silver colloids [33,34]. 

Figure 2b illustrates the UV-visible absorption spectra of initial fipronil (100 ppm), pristine AgNPs, AgNPs-CTAC, and fipronil on CTAC-coated AgNPs. The UV-visible absorption spectra of fipronil and AgNPs demonstrated maximum values centered at about 280 and 420 nm, respectively. The addition of fipronil into the AgNPs solution resulted in the appearance of peaks at around 280 and 420 nm, indicating the substantial adsorption of fipronil on the AgNPs surface. The absorbance intensity peak, at 280 nm, of the overlapping spectrum of AgNPs-CTAC-fipronil was lower than the initial fipronil spectra. The unbound fipronil could be estimated by performing the following procedure of the incubation with AgNPs-CTAC-fipronil, the subsequent centrifugation, and the removal of the precipitate AgNPs-CTAC-fipronil. As illustrated in Figure 2b, the loading efficiency [29] of fipronil on AgNPs-CTAC was calculated using the UV-visible absorption measurements from the initial free amount of fipronil subtracted by the supernatant solution of the excessive fipronil on the basis of the following equation, where O.D. stands for optical density.
$$\mathrm{Fipronil}\;\mathrm{Loading}\;\mathrm{efficiency}\;\left(\%,\frac{w}{w}\right)=\frac{\mathrm{Free}\;\mathrm{Fipronil}\;O.D-\mathrm{Supernatant}\;\mathrm{of}\;\mathrm{Fipronil}\;O.D}{\mathrm{Free}\;\mathrm{Fipronil}\;O.D}\times 100\%$$

The loading efficiency of fipronil on the AgNPs-CTAC was estimated to be ~38.09%. These UV-visible results were also consistent with the zeta-potential data, which indicated that the average charge surface of AgNPs and CTAC-capped AgNPs were at −48.82 ± 3.29 and 46.97 ± 2.14 mV, respectively (Figure 2d). The treatment with CTAC incompletely prevented the extensive aggregation of AgNPs, considering the slightly increased average diameter of AgNPs and CTAC-capped AgNPs of ∼50 and ∼88 nm, respectively (Figure 2c).

In addition, we investigated the effect of laser wavelength on the SERS of a fipronil-metal complex. The visible lasers at both 514 and 633 nm excitations were used to obtain the SERS spectra of fipronil on AgNPs. These results fundamentally validate the relationship between the SERS spectra and the energy of laser wavelength excitation at 514 and 633 nm, suggesting that the higher energy of the excitation wavelength at 514 nm induces strong SERS signals of fipronil on Ag surfaces, which would be unfavorable for the lower energy of the excitation wavelength at 633 nm, as well as at 780 nm from the previous report [5]. As shown in Figure 3b, our method was able to detect fipronil by both the Raman spectra and SERS approaches. The SERS method demonstrated higher selective and sensitive ability than the Raman spectroscopy method in much of the previously reported works [11,12,13,14,15,16,17,18,19,20,21,22].

CTAC shows an extremely weak SERS spectrum on AgNPs and there is almost no SERS signal in our measurement. Although not shown, SERS experiments with AuNPs-fipronil (with lasers at 514 and 633 nm) and AgNPs-fipronil (with lasers at 633 nm) were performed. Despite extensive efforts to find the Raman peaks of fipronil, we were unable to observe any strong bands that could be ascribed to the adsorbates. According to our accumulated data, SERS of fipronil could be achieved by using AgNPs with lasers at 514 nm. The representative SERS spectra of AgNPs-fipronil are shown in Figure 3b. From the spectra, we can see the strongest characteristic peak at about 2236 cm^−1^, which was likely produced from the nitrile (−C≡N) group. This position is unique and can be differentiated from many other analytes that do not contain the nitrile functional group. This peak shift was ideal for quantitative determination. Other Raman shifts are correlated with δ(C=O), δ(CH), and δ(CF) bending vibrations, as well as ν(CC), ν(CN), and ν(S=O) stretching vibrations. The DFT results of the SERS spectra simulations of Ag_6_-fipronil and Au_6_-fipronil also show, similarly, their strongest Raman peaks being characteristic of a nitrile group (as shown in Figure 3a).

A concentration range from 0.0001 to 200 ppm was tested for linearity and limit of detection (LOD) of fipronil. The high-concentration fipronil solution was used as a standard solution to obtain nine concentrations of 0.0001, 0.001, 0.01, 0.1, 1, 10, 50, 100, and 200 ppm. Figure 4a shows that the SERS signal intensity at 2236 cm^−1^ was positively correlated with concentrations of fipronil. This allowed us to experimentally establish the standard curve toward fipronil. Based on the 2236 cm^−1^ band, the detection limit of fipronil was then calculated. This calculation was approximated from the linear fitting of the calibration curve equation as:
Y (2236 cm^−1^ band intensities) = B (intercept) + A (slope) × X (fipronil concentrations)

LOD can be statistically determined from the slope of the calibration curve and is generally defined as LOD = 3 × SD ÷ slope, where SD is the standard deviation of the noise. The r^2^ of the standard curve is 0.985 and the calculated LOD in the standard curve is 0.001 ppm. Although the LOD for fipronil is not as low as the LC-MS/MS method [2,6,7], this approach can detect fipronil after isolating it from food, such as by ethanol, acetone, or other organic solvents. The application of this method can therefore be adjusted for many fields, such as food science processing and food, environment, or agriculture safety.

As shown in Table 2, the DFT has also been enforced to approximate the fipronil energetic stabilities on six Au atom clusters and six Ag atom clusters. Remarkably, the energetically optimized structural representations of fipronil in close proximity to the Ag_6_ cluster at the nitrogen atom of pyrazole and the CN group were slightly different. Theoretical results have shown that the value of the energy binding of the fipronil molecule on the Ag_6_ cluster was not greatly dependent on different binding positions.

## 4. Conclusions

Herein, we reported an ultrasensitive SERS method for fipronil detection using AgNPs with a laser excitation at 514 nm. The proposed procedure was demonstrated as a potentially convenient method for special detection of fipronil with a LOD of 0.001 ppm. In addition, the combination of SERS experiments with DFT calculations may be a novel fipronil detection method in food science that is low cost and easy to use. This method is expected to offer significant advantages over Raman spectroscopy due to the improvements produced by nano-roughened metallic surfaces.

## Figures and Tables

**Figure 1 sensors-19-01355-f001:**
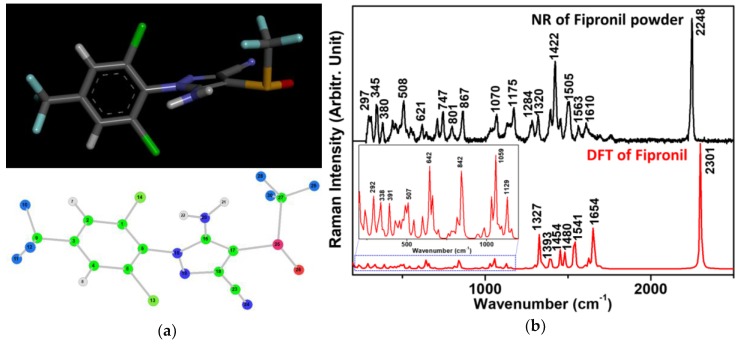
(**a**) Fipronil structure and numbered fipronil structure; (**b**) experimentally measured and theoretically calculated Raman vibrations of fipronil, respectively. The normal Raman (NR) spectrum was baseline corrected for the standard fipronil powder (at 633 nm).

**Figure 2 sensors-19-01355-f002:**
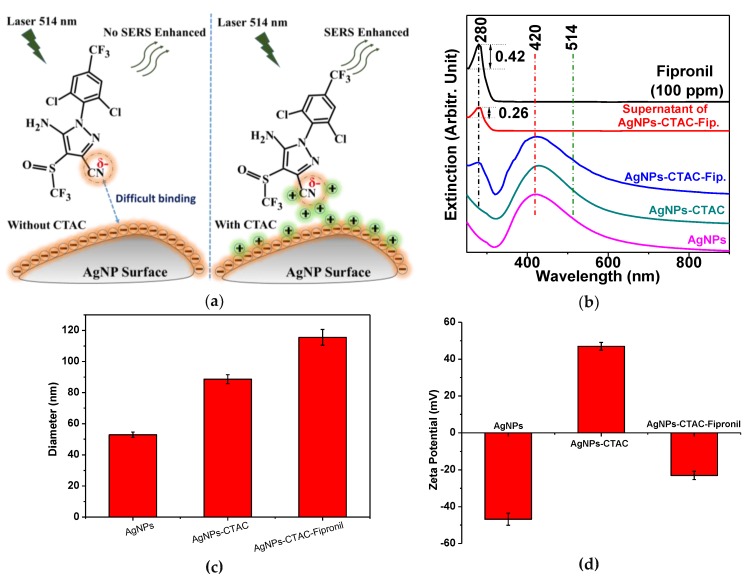
(**a**) Plausible adsorption mechanism of fipronil on AgNP surfaces; (**b**) UV-visible absorption spectra of initial fipronil (100 ppm), the supernatant solution of the excessive fipronil obtained by the reaction with CTAC-capped AgNPs, the subsequent centrifugation, and the removal of the supernatant fipronil from the precipitate AgNPs-CTAC-fipronil. The absorption values were compared to yield the loading efficiency to estimate the amount of the adsorption of fipronil on AgNPs; (**c**) the size distribution data are shown of initial AgNPs (∼50 nm), AgNPs-CTAC (∼88 nm), and AgNPs-CTAC-fipronil (∼116 nm); (**d**) surface charge changes after the adsorption of CTAC and subsequently fipronil on AgNPs. Error bars showed the standard deviation after the three repetitive measurements.

**Figure 3 sensors-19-01355-f003:**
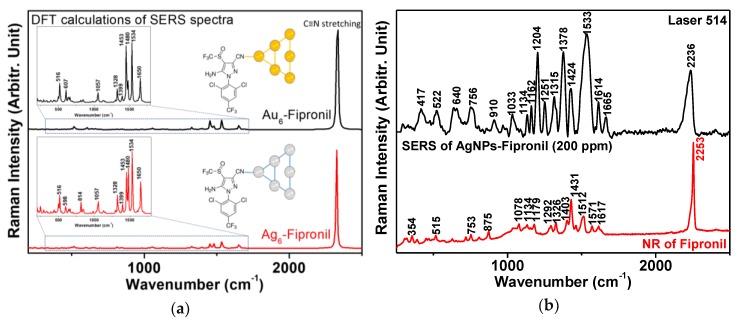
(**a**) Theoretically estimated Raman vibrational spectra of fipronil binding at the nitrile moiety on Au_6_ and Ag_6_ atom clusters; (**b**) experimentally recorded SERS vibrational spectra of AgNPs-fipronil and NR spectrum of fipronil powder (laser at 514 nm).

**Figure 4 sensors-19-01355-f004:**
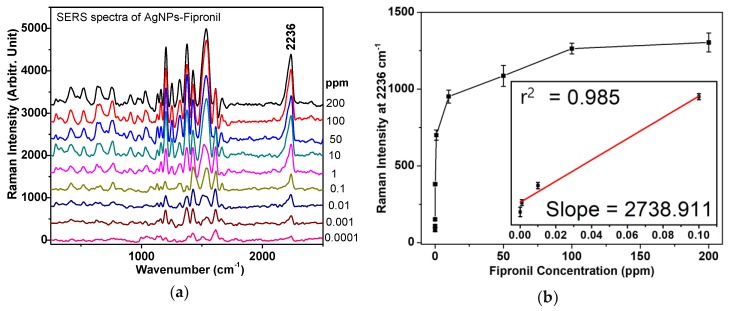
(**a**) Fipronil concentration-dependent SERS spectra using AgNPs (with the irradiation at 514 nm) in the range of 0.0001–200 ppm; (**b**) The calibration curves of the vibrational band intensities at ~2236 cm^−1^, the inset shows fitting of the linear region of 0.0001–0.1 ppm. Each SERS experiment was repeated three times independently to ensure the reliability of this evaluation. The mean and standard deviation values were defined.

**Table 1 sensors-19-01355-t001:** Spectral data and vibrational assignments for theoretical density functional theory (DFT) calculation of fipronil; experimental measurement of the NR of fipronil powder (at 633 nm and 514 nm) and SERS of AgNPs-fipronil (at 514 nm), respectively.

^a^ DFT	NR 633 nm	NR 514 nm	SERS 514 nm	^b^ Assignments Based on PED Calculations
2301	2248	2253	2236	ν(N_24_-C_23_) (85%) + ν(C_23_-C_18_) (15%)
1654			1665	ν(C_1_-C_2_) (26%) + ν(C_5_-C_4_) (12%) + ν(C_3_-C_2_) (10%) + ν(C_4_-C_3_) (12%)
	1610	1617	1614	ν(C_17_-C_16_) (32%) + ν(N_20_-C_16_) (35%) + β(H_22_-N_20_-H_21_) (14%)
1541	1563	1571	1533	ν(N_15_-C_16_) (10%) + ν(N_15_-C_6_) (29%)
1480	1505	1512		ν(C_17_-C_16_) (10%) + ν(N_19_-C_18_) (20%)
1454		1431		ν(C_23_-C_18_) (21%) + β(C_18_-N_19_-N_15_) (20%) + β(C_17_-C_16_-N_15_) (13%) + β(C_16_-N_15_-N_19_) (13%)
	1422	1403	1424	ν(C_1_-C_2_) (15%) + ν(C_5_-C_4_) (15%) + β(H_7_-C_2_-C_3_) (20%) + β(H_8_-C_4_-C_5_) (17%)
1393			1378	ν(N_19_-C_18_) (34%) + β(C_18_-N_19_-N_15_) (11%) + β(C_16_-N_15_-N_19_) (11%)
1327	1320	1326	1315	ν(N_19_-C_18_) (13%) + ν(C_9_-C_3_) (15%)
	1284	1292	1251	ν(C_9_-C_3_ )(23%) + β(H_7_-C_2_-C_3_) (11%) + β(H_8_-C_4_-C_5_) (18%)
	1175	1179	1204	ν(C_17_-C_16_) (10%)
1129		1134	1134	ν(F_29_-C_27_) (34%) + ν(F_30_-C_27_) (24%) + ν(F_28_-C_27_) (14%) + β(F_30_-C_27_-F_29_) (10%)
1059	1070	1078	1033	ν(C_23_-C_18_) (14%) + β(C_17_-C_16_-N_15_) (13%) + β(C_16_-N_15_-N_19_) (13%)
842	867	875	910	β(C_4_-C_3_-C_2_) (18%)
	801			ν(S_25_-O_26_) (89%)
	747		756	δ(C_18_-N_19_-N_15_-C_6_) (14%) + γ(C_23_-C_17_-N_19_-C_18_) (23%) + ν(C_3_-C_2_) (23%)
642	621		640	ν(F_29_-C_27_) (11%) + ν(F_30_-C_27_) (13%) + ν(F_28_-C_27_) (14%) + β(F_29_-C_27_-F_28_) (15%) + β(F_30_-C_27_-F_29_) (20%) + β(F_28_-C_27_-F_30_) (15%)
507	508	515	522	ν(C_23_-C_18_) (15%) + β(C_18_-N_19_-N_15_) (23%)
391	380		417	β(N_20_-C_16_-C_17_) (12%) + β(F_10_-C_9_-F_12_) (17%)
338	345	354		δ(N_24_-C_23_-C_18_-C_17_) (14%) + δ(C_17_-C_16_-N_15_-C_6_) (10%) + δ(C_18_-N_19_-N_15_-C_6_) (12%) + ν(C_3_-C_2_) (21%)
292	297			β(F_12_-C_9_-F_11_) (20%) + γ(F_10_-C_3_-F_12_-C_9_) (17%)

^a^ No scale factor was applied. ^b^ Abbreviations: δ, Torsion; ν, stretching; β, in-plane bending; γ, out-of-plane bending. Unit in cm^−1^. Atomic numbering depicted in Figure 1a.

**Table 2 sensors-19-01355-t002:** Relative energetic stabilities of fipronil binding with Au_6_ cluster and Ag_6_ cluster.

Binding Positions	Energy (a.u.)	Differences (a.u.)	Differences
Ag_6_-Fironil (binding at pyrazole)	−2267.49982461	0.00305258	1.915523 kcal/mol
Ag_6_-Fironil (binding at CN)	−2267.50287719	0.00000000	
Au_6_-Fironil (binding at pyrazole)	−2205.66239617	0.00220096	1.381123 kcal/mol
Au_6_-Fironil (binding at CN)	−2205.66459713	0.00000000	

Unit in a.u. (hartree).

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
