# Peer review of "Surface-Enhanced Raman Scattering Detection of Fipronil Pesticide Adsorbed on Silver Nanoparticles"

_sensors, 2019, doi:10.3390/s19061355_

Round 1

Reviewer 1 Report

In this paper, a method for the SERS detection of fipronil (a pesticide) is presented. Different amounts of fipronil are mixed with silver nanoparticles synthesized by citrate reduction and a cationic surfactant: the linearity range and the limit of detection are determined. DFT calculations of the fipronil molecule in vacuum and bound to gold and silver clusters with 6 atoms are carried out and compared to experiments.

Overall, I think that this paper is not suitable for publication in Sensors. The SERS detection of pesticides as such is an interesting subject but the original elements contained in this paper are not enough for a publication and the scientific approach the authors adopted is not sufficiently sound.

More specifically:

-     When DFT calculations are carried out to reproduce the SERS spectra of a molecule adsorbed on a surface, several cluster sizes (also with different charges) are normally simulated to find out the one that better reproduces the experiments (10.1021/jp073914h, 10.1103/PhysRevLett.106.083003). Here the authors used only a neutral cluster with 6 atoms, saying that this is “one of the simplest model available” (line 101). Moreover, it is good practice to employ different functionals and basis sets to confirm that the simulations do not depend on the method used: here, only one functional and one basis set have been used. In figure 1b, DFT and experimental spectra  are reported: below 1000 cm-1 they are grossly different and no comments on this point are given.

- CTAC has as a counter ion Cl-. It is well known the activation effect of this ion on the SERS spectra of some molecules (doi:10.1016/j.molstruc.2003.07.026; doi:10.1002/(SICI)1097-4555(199602)27:2<105::AID-JRS933>3.0.CO;2-L) and its presence may also influence the charge of the silver clusters used in simulations: some comments and in depth studies on this point should have been reported.

-  The model proposed in Figure 2b is inconsistent with the calculations: in Figure 2b the authors suppose that CTAC acts as an electrostatic linker between the nanoparticle and the fipronil, while in the simulated spectra the authors assume that the molecule binds to the silver surface.

- At line 175 it is written that “The addition of fipronil into the AgNPs solution resulted in the appearance of peaks at around 280 and 420 nm, indicating the substantial adsorption of fipronil on the AgNPs surface.” Why? Also in the case in which fipronil is not adsorbed on silver nanoparticles the peaks at 280 and 420 nm would be present in the spectrum.

-   A control spectrum of CTAC should have been reported.

Author Response

Reply to Reviewer #1's Comments

The authors deeply appreciate the Reviewer #1 for giving helpful comments. We performed the additional experiments to address the issues raised by the Reviewer #1 and corrected unclear expressions and errors in the manuscript. According to Reviewer #1’s comments, we revised the manuscript as follows:

In this paper, a method for the SERS detection of fipronil (a pesticide) is presented. Different amounts of fipronil are mixed with silver nanoparticles synthesized by citrate reduction and a cationic surfactant: the linearity range and the limit of detection are determined. DFT calculations of the fipronil molecule in a vacuum and bound to gold and silver clusters with 6 atoms are carried out and compared to experiments.

Overall, I think that this paper is not suitable for publication in Sensors. The SERS detection of pesticides as such is an interesting subject but the original elements contained in this paper are not enough for a publication and the scientific approach the authors adopted is not sufficiently sound.

More specifically:

1) When DFT calculations are carried out to reproduce the SERS spectra of a molecule adsorbed on a surface, several cluster sizes (also with different charges) are normally simulated to find out the one that better reproduces the experiments (10.1021/jp073914h, 10.1103/PhysRevLett.106.083003). Here the authors used only a neutral cluster with 6 atoms, saying that this is “one of the simplest models available” (line 101). Moreover, it is good practice to employ different functionals and basis sets to confirm that the simulations do not depend on the method used: here, only one functional and one basis set have been used. In figure 1b, DFT and experimental spectra are reported: below 1000 cm-1 they are grossly different and no comments on this point are given.

® Thank you for your comments. We have performed additional DFT calculations of normal and SERS spectra of the free and the adsorbed fipronil on Ag surfaces consisted of several cluster sizes and different charges. In the case of normal Raman spectra, we newly performed the three different basis sets of B3LYP/6-311G(d,p), B3LYP/cc-pVTZ, and B3LYP/6-31G(d,p) were performed. No scale factor was applied to compare the peak positions of the simulated and experimental normal Raman spectra. On the basis of the spectral positions of the nitrile (−C≡N) characteristic peaks and the numerous bands between 200-1800 cm-1, the functional B3LYP/6-31G(d,p) was chosen, since it showed the best match with the experimental values, whereas the B3LYP/LANL2DZ basis set was generally employed for the calculations of surface complexes constituted by organic ligands bound to Ag atoms. In addition to the six neutral cluster Ag atoms, we newly tested three and four atoms with the positive charge. Our results indicated that Ag6 neutral cluster-fipronil (binding at CN) would show the best match in comparison to SERS experiments.

Figure S1. Simulated spectra of the free and adsorbed states of fipronil at the different functionals in the DFT calculations.

 In order to address the raised issue by the Reviewer #1, we newly added an inset of the 200-1200 cm-1 region in Figure 1b to better compare experimental and calculational spectra below 1000 cm-1.

Figure 1b. Experimentally measured and theoretically calculated Raman vibrations of fipronil, respectively. The baseline correction for normal Raman (NR) spectrum was recorded for the standard fipronil powder (laser 633 nm).

 2) CTAC has as a counter ion Cl-. It is well known the activation effect of this ion on the SERS spectra of some molecules (doi:10.1016/j.molstruc.2003.07.026; doi:10.1002/(SICI)1097-4555(199602)27:2<105::AID-JRS933>3.0.CO;2-L) and its presence may also influence the charge of the silver clusters used in simulations: some comments and in-depth studies on this point should have been reported.

® Thank you for your comments. We newly included the two articles and added the following sentences in the revised manuscript as follows:

Moreover, CTAC has a counter ion chloride in molecular and chloride-capped AgNPs allow for high-intensity SERS spectra of molecules to be obtained, it may also play a key role in understanding some fundamental principles behind SERS. Chloride was well known the chloride activation effect on the SERS spectra of some molecules such as phthalazine and pyridine adsorbed on silver colloids [32,33].

32. Muniz‐Miranda, M.; Sbrana, G. Evidence for surface Ag+ complex formation by an anion‐induced effect in the SER Spectra of phthalazine adsorbed on silver sols. J. Raman Spectrosc. 1996, 27, 105-110.

33. Otto, A.; Bruckbauer, A.; Chen, Y. X. On the chloride activation in SERS and single molecule SERS. J. Mol. Struct. 2003, 661, 501-514.

3) The model proposed in Figure 2b is inconsistent with the calculations: in Figure 2b the authors suppose that CTAC acts as an electrostatic linker between the nanoparticle and the fipronil, while in the simulated spectra the authors assume that the molecule binds to the silver surface.

® Thank you for your comments. Since CTAC shows an extremely weak SERS spectrum on AgNPs, it did not affect SERS signals in our measurements. Therefore, in the simulated spectra, we assume that the molecule binds to the silver surface, despite the presence of CTAC in experimental SERS measurement.

4) At line 175 it is written that “The addition of fipronil into the AgNPs solution resulted in the appearance of peaks at around 280 and 420 nm, indicating the substantial adsorption of fipronil on the AgNPs surface.” Why? Also, in the case in which fipronil is not adsorbed on silver nanoparticles the peaks at 280 and 420 nm would be present in the spectrum.

® Thank you for your comments. In order to address the Reviewer #1’s comments on the mechanism of the adsorption of fipronil on silver, we newly included the loading efficiency in the revised manuscript.

We removed the unattached fipronil by centrifugation and measured the loading efficiency of fipronil on Ag on the basis of the previous report [35]. The following sentences were added for better presentation.

As illustrated in Figure 2b, the loading efficiency [35] of fipronil on AgNPs-CTAC was calculated using the UV-Vis absorption measurements from the initial free amount of fipronil subtracted by the supernatant solution of the excessive fipronil obtaining by incubation with AgNPs-CTAC then centrifugation and removal of precipitate AgNPs-CTAC-Fipronil on the basis of the following equation, where O.D. stands for optical density.

The loading efficiency of fipronil on the AgNPs-CTAC was estimated to be ~38.09%.

Figure 2b. UV–visible extinction spectra of initial fipronil (100 ppm), the supernatant solution of the excessive fipronil obtained by reaction with AgNPs-CTAC, subsequent centrifugation. and the removal of the supernatant fipronil from the precipitate AgNPs-CTAC-Fipronil. The extinction values were compared to yield the loading efficiency to estimate the amount of the adsorption of fipronil on AgNPs.

5) A control spectrum of CTAC should have been reported.
® We newly measured the control spectrum of CTAC to reveal weak features as shown in Figure S2.

Figure S2. Raman spectrum of CTAC on AgNPs.

Finally, authors would like to appreciate the Reviewer #1 once again for giving valuable comments. Thank you.

Reviewer 2 Report

Manuscript ID: sensors-440039

Title:  Surface-enhanced Raman scattering detection of fipronil pesticide adsorbed on silver nanoparticles

Authors: Ly et al.

The work presented in the manuscript is certainly valuable, reporting original data related to the detection and determination of concentration of fipronil, by using the SERS technique couplet to DFT quantum chemical calculations. The subject of the manuscript is well within the scope of the journal and the results are original.

However, a revision is recommended before publication, according to the following comments:

1. In the Abstract section, the word "method" should be removed from the line 12 and the word "spectroscopy" in line 18 should be replaced by "spectrum".

It is not clear what the authors want to say with "of a molecular form" in line 13.

2. Section 2.3

a. My suggestion is to replace the word "established" in line 96 by "performed".

b. Also, "using the B3LYP/6-31G** basis set in the free molecule" should be replaced by "at the B3LYP/6-31G(d,p) level of theory on the free molecule".

The 6-31G** notation for a basis set is old-fashioned and it does not tell exactly what kind and how many polarization basis functions have been used for the heavy and H atoms. Thus, 6-31G** could point to the 6-31G(d,p) basis set but also to the 6-31G(2df,p) basis set, for instance.

c. Starting with line 100, the authors say that the Ag_6 cluster was optimized prior to binding with fipronil molecule. This statement leads to the conclusion that the whole complex Ag cluster+fipronil was not fully optimized. The authors must clarify this aspect.

3. Table 1.

a. The band observed in the Raman spectrum at 1693 cm^-1 (excitation 633 nm) is very weak and I have doubts that it could be assigned to a beta(NH_2)+niu(CN) normal mode.

b. Moreover, there are (too) large differences between the wavenumbers assigned to the same vibration, obtained with the two excitation lines, (see for instance the bands at 1070 and 1134 cm^-1.

Section 3.2

a. On page 10, line 201, my recommendation is to replace "the previous research" by "the previously reported works".

b. Page 7, line 215, the symbol delta must be replaced by niu.

c. For an easier comparison of the SERS spectra, I would recommend to couple Fig. 3.b with Fig.1.b.

Conclusion section

Line 253: The recommendation is to replace "irradiation" by "excitation line of".

Finally, in order to be fair in the treatment of previous literature related to the detection of very low concentration of hazardous compounds, the authors could consider the work by Pinzaru et al. (Journal of Raman Spectroscopy, 2016, 47, 636–642).

Author Response

Reply to Reviewer #2's Comments

The authors deeply appreciate the Reviewer #2 for giving positive comments. We performed the new experiments to address the issues raised by the Reviewer #2 and corrected unclear expressions and errors in the manuscript. According to Reviewer #2’s comments, we revised the manuscript as follows:

The work presented in the manuscript is certainly valuable, reporting original data related to the detection and determination of the concentration of fipronil, by using the SERS technique couplet to DFT quantum chemical calculations. The subject of the manuscript is well within the scope of the journal and the results are original. However, a revision is recommended before publication, according to the following comments.

1) In the Abstract section, the word "method" should be removed from line 12 and the word "spectroscopy" in line 18 should be replaced by "spectrum". It is not clear what the authors want to say with "of a molecular form" in line 13.

® We corrected in the manuscript as follows:

In line 12: we removed the word “method”.

In line 13: we changed from “of a molecular form” to “of a molecular”.

In line 18: we replaced the word “spectroscopy” by “spectrum”.

2) Section 2.3

a. My suggestion is to replace the word "established" in line 96 by "performed".

® Thank you for your comments. We replaced the word “established” by “performed” in the manuscript.

b. Also, "using the B3LYP/6-31G** basis set in the free molecule" should be replaced by "at the B3LYP/6-31G(d,p) level of theory on the free molecule". The 6-31G** notation for a basis set is old-fashioned and it does not tell exactly what kind and how many polarization basis functions have been used for the heavy and H atoms. Thus, 6-31G** could point to the 6-31G(d,p) basis set but also to the 6-31G(2df,p) basis set, for instance.

® We replaced “using the B3LYP/6-31G** basis set in the free molecule” by “at the B3LYP/6-31G(d,p) level of theory on the free molecule” in the manuscript.

c. Starting with line 100, the authors say that the Ag6 cluster was optimized prior to binding with fipronil molecule. This statement leads to the conclusion that the whole complex Ag cluster+fipronil was not fully optimized. The authors must clarify this aspect.

® We corrected the manuscript as follows:

“AgNP modeling was conducted using the B3LYP/LANL2DZ method to optimize Ag6 cluster geometry prior to binding with the fipronil molecule, then the whole complex of Ag6-fipronil was fully optimized again.”

3) Table 1.

a. The band observed in the Raman spectrum at 1693 cm-1 (excitation 633 nm) is very weak and I have doubts that it could be assigned to a beta(NH2)+niu(CN) normal mode.

® We omitted this peak in the revised manuscript for better presentation.

b. Moreover, there are (too) large differences between the wavenumbers assigned to the same vibration, obtained with the two excitation lines, (see for instance the bands at 1070 and 1134 cm-1.

® We attempted to improve the large differences as newly replaced in Figure 3 and 1 of the revised manuscript.

4) Section 3.2

a. On page 10, line 201, my recommendation is to replace "the previous research" by "the previously reported works".

® We replaced “the previous research” by “the previously reported works” in the manuscript.

b. Page 7, line 215, the symbol delta must be replaced by nu.

® We corrected from the symbol “delta” to “nu” in the revised manuscript.

c. For an easier comparison of the SERS spectra, I would recommend to couple Fig. 3.b with Fig.1.b.

® Thank you for your comments. After consideration of this comment, we have remained Figure 3 and 1 to discuss our spectral analysis from the normal spectra to the SERS spectra.

5) Conclusion section

Line 253: The recommendation is to replace "irradiation" by "excitation line of".

® We replaced “irradiation” by “excitation line of” in the manuscript.

6) Finally, in order to be fair in the treatment of previous literature related to the detection of very low concentration of hazardous compounds, the authors could consider the work by Pinzaru et al. (Journal of Raman Spectroscopy, 2016, 47, 636–642).

® We newly included the reference as [12]. Thank you.

12.        Pinzaru, S. C.; Müller, C.; Tódor, I. S.; Glamuzina, B.; Chis, V. NIR-Raman spectrum and DFT calculations of okadaic acid DSP marine biotoxin microprobe. J. Raman Spectrosc. 2016, 47, 636-642.

Finally, authors would like to appreciate the Reviewer #2 once again for giving valuable comments. Thank you.

Reviewer 3 Report

Recommendation – Publish with minor revisions

Joo et al., used AgNPs and DFT calculations to examine fipronil using SERS measurements. The manuscript can be accepted after minor revision.

Comments

1.      Can the authors provide the SEM images of the metal nanoparticles?

2.      What are the relative standard deviation and enhancement factor of the SERS substrate and measurements?

3.      The authors can mention the laser power and microscope objective in the instrumentation section.

4.      Did the authors observe any Raman signals from CTAC and citrate molecules? Citrate molecules would exhibit SERS signals especially at lower concentrations of target molecules.  

Author Response

Reply to Reviewer #3's Comments

The authors deeply appreciate the Reviewer #3 for giving positive comments. We performed the new experiments to address the issues raised by the Reviewer #3 and corrected unclear expressions and errors in the manuscript. According to Reviewer #3’s comments, we revised the manuscript as follows:

Joo et al., used AgNPs and DFT calculations to examine fipronil using SERS measurements. The manuscript can be accepted after minor revision.

Comments:

1) Can the authors provide the SEM images of the metal nanoparticles?

® Thank you for your comments. We provided TEM images of AgNPs as revealed in Figure S3.

                                               Figure S3. Electron microscopic images of AgNPs.

 2) What is the relative standard deviation and an enhancement factor of the SERS substrate and measurements?

® As listed in Figure 4(b), the standard deviations of the SERS measurements were measured to be 2.01-14.3%. The enhancement factor was estimated to be 3.7 x 103.

3) The authors can mention the laser power and microscope objective in the instrumentation section.

® Thank you for your comments. We provided it in the manuscript as follows:

“All Raman data was obtained using either a Renishaw RM 1000 spectrometer Raman microscope system with a 633 nm (25 mW) HeNe excitation laser or a 514 nm (195 mW) Melles Griot Model 35-LAP-431-220 multiline Argon ion laser and a CCD camera. The SERS spectra were recorded via spectroscopic glass tubes after the preparation of the fipronil-metal samples with an x20 objective.”

4) Did the authors observe any Raman signals from CTAC and citrate molecules? Citrate molecules would exhibit SERS signals especially at lower concentrations of target molecules.

® Thank you for your comments. We mentioned it in the manuscript as follows:

“CTAC shows an extremely weak SERS spectrum on AgNPs to obtain almost no SERS signal in our measurement.”

Figure S2. Raman spectrum of CTAC on AgNPs.

 Finally, authors would like to appreciate the Reviewer #3 once again for giving valuable comments. Thank you.

Reviewer 4 Report

Surface-enhanced Raman scattering detection of fipronil pesticide adsorbed on silver nanoparticles The submitted manuscript titled “Surface-enhanced Raman scattering detection of fipronil pesticide adsorbed on silver nanoparticles” presented a simple approach for detection of a known pesticide using silver nanoparticles and surface enhanced Raman spectroscopy. The authors further corroborated their signal detection of fipronil by employing density functional theory calculations. The manuscript and experimental techniques presented in the manuscript to achieve signal detection of fipronil appear appropriate for the journal Sensors. However, several concerns (listed below) in this work require major revision before the manuscript should be considered for publication. General Concerns 1. Page 2: Layout of the introduction is not easy to follow. It is recommended the paragraphs introducing fipronil (starting on page 2 line 54) be moved to the beginning of the introduction, i.e. ahead of how the authors plan to study fipronil with SERS and DFT. Then the reader will have an idea of fipronil and its importance prior to discussion of its detection with SERS. As the manuscript is currently, the reader is introduced to Raman, DFT and SERS of fipronil before learning what fipronil is and why it should be studied. 2. Page 1 Lines 31-33: Authors previously suggested that “SERS procedures for the detection of fipronil are unavailable in the literature” however they then go on to detail and cite a case where SERS is employed to detect fipronil in eggshells and liquid eggs (Page 2 Lines 48-50). These statements are a contradiction and should be edited. 3. SERS based detection of pesticides are well reported in the literature and it is not fully clear what the novelty of the present manuscript is. While the merit of reporting specific details of fipronil detection with a SERS technique and analysis with DFT is certainly relevant, it is unclear whether the SERS protocol in the present manuscript has been developed by others in the literature or if it is new as suggested when the authors state “our method.” How does this SERS based protocol for fipronil detection compare and contrast to those already in the literature employing similar types of SERS detection? Specifically has a similar SERS protocol been successfully employed with other pesticides, e.g. template-free AgNPs with CTAC? It is highly recommended the authors expand their introduction to include these items with comparisons to literature of other pesticides. 4. The authors quote the maximum residue limit of 0.01 ppm for fipronil in food set in the U.S. and Europe, however values range depending on the type of tissue being sold as food. Can the authors clarify this point in the text? 5. How does the Raman spectrum collected compare to the literature reports of Raman detection of fipronil in or on eggs (ref 17)? 6. The authors suggest they attempted SERS experiments with AuNPs but were unable to observe any strong bands. The authors make no mention regarding the experimental materials or procedural details of creating their AuNPs nor do they report the size or the position of the UVvis band. The authors must report these details of the AuNPs if a qualitative comparison is to be drawn. What are some possible reasons why AuNPs failed fipronil detection? Can these same problems occur with AgNPs leading to poor reliability in fipronil detection? Further justification of why AuNPs did not detect fipronil is required. 7. Page 5: Line 175-7: Authors state that there is “substantial adsorption of fipronil on the AgNPs surface. What evidence is there from the UV-vis spectra (Figure 2b) that fipronil is adsorbed onto the Ag nanoparticles and not just present in solution? Would a spectrum of CTAC-Fip (absent AgNPs) not also result in a band positioned at 280nm? The authors state there is an appearance of the 280 nm band in the AgNPs-CTAC-Fip spectrum suggesting fipronil is present in the mixture’s solution, but without further evidence (e.g. a shift in the 280nm band position) it is difficult to support the statement that fipronil is adsorbed on silver. 8. UV-vis spectra should have the y-axis labelled as extinction. Extinction is the sum of both absorbance and scattering processes. The scattering component is often overlooked or ignored in the literature however the scattering process may still be significant here as observed in the UVvis of the AgNPs (Figure 2b), therefore this plot would be more correctly termed extinction in the figure and the text. 9. Can the authors expand on the comment of testing for linearity, as it is an important parameter for determining the useful range of this technique? More generally, can the authors comment on the applicability of fitting the Raman intensity versus concentration to a linear line? Some other SERS based platforms for pesticide detection have reported linearity with double logarithmic scales (Freundlich adsorption of a solute on the surface of an adsorbent). The inset used by the authors to fit a linear line still appears to have curvature as predicted in a Freundlich adsorption isotherm. 10. Can the authors include the value of the slope calculated in Figure 4B? Responsivity, or slope of a calibration curve, for a specific protocol is a useful and therefore relevant parameter that should be reported in sensing. 11. The authors conclude in the main text that an energy difference of ~2 kcal/mol between binding at the nitrile and the N1 position to the Ag cluster is “negligible,” while in the abstract they suggest this same difference suggests binding through the nitrile is favourable, i.e. not negligible. Can the authors elaborate on these statements and the differences between the nitrile and N1 position? Further how was binding between the fipronil and the Ag cluster set or optimized in the DFT calculations prior to calculation of the Raman spectra? The experimental details suggest the cluster was optimized separately from the free molecule, however would interaction of the cluster and free molecule impact the optimized structures? Determining how fipronil is optimized with a Ag cluster will ultimately impact the resulting Raman spectrum calculated and therefore it is recommended the authors comment on this detail. 12. The bands in Figure 3a are difficult to distinguish due to their relative size compared to the nitrile band. Can an inset of the 400-1700 cm-1 regions be included? Can DFT predict a noticeable shift in the nitrile fipronil band upon inclusion of the Ag6 cluster relative to the free molecule? Can the shift if the vibrational mode of the nitrile group be correlated with interaction with the AgNPs surface? Specific Concerns 1. Is the experimental NR spectrum in Figure 1b baselined corrected? If so this should be mentioned in the caption. 2. Page 5 Line 152: Wrong units stated for Raman Shift “Unit in cm1” should be corrected to cm-1. 3. Figure 3b appears to show the molecular structure of the fipronil bound to the metal clusters through the nitrile moiety, in opposition to the stated structure in the caption as bound through the N1 atom. The authors must clarify.

Author Response

Reply to Reviewer #4's Comments

The authors deeply appreciate the Reviewer #4 for giving helpful comments. We performed the new experiments to address the issues raised by the reviewers and corrected unclear expressions and errors in the manuscript. According to Reviewer #4’s comments, we revised the manuscript as follows:

Surface-enhanced Raman scattering detection of fipronil pesticide adsorbed on silver nanoparticles. The submitted manuscript titled “Surface-enhanced Raman scattering detection of fipronil pesticide adsorbed on silver nanoparticles” presented a simple approach for detection of a known pesticide using silver nanoparticles and surface-enhanced Raman spectroscopy. The authors further corroborated their signal detection of fipronil by employing density functional theory calculations. The manuscript and experimental techniques presented in the manuscript to achieve signal detection of fipronil appear appropriate for the journal Sensors. However, several concerns (listed below) in this work require major revision before the manuscript should be considered for publication.

General Concerns:

1) Page 2: Layout of the introduction is not easy to follow. It is recommended the paragraphs introducing fipronil (starting on page 2 line 54) be moved to the beginning of the introduction, i.e. ahead of how the authors plan to study fipronil with SERS and DFT. Then the reader will have an idea of fipronil and its importance prior to discussion of its detection with SERS. As the manuscript is currently, the reader is introduced to Raman, DFT, and SERS of fipronil before learning what fipronil is and why it should be studied.

® We arranged and rephrased the introduction part again in the manuscript as follows:

1. Introduction

Fipronil, (5-amino-1-[2,6-dichloro-4-(trifluoromethyl) phenyl]-4-[(trifluoromethyl)-sulfinyl]-1H-pyrazole-3-carbonitrile), is a potent insecticide of the phenylpyrazole group that is widely used in agriculture to control pests with high efficiency at very low concentrations [1]. Fipronil can impact γ-aminobutyric acid reception in nerve transmission and effectively block γ-aminobutyric acid–regulated chloride channels in the nervous system, paralyzing or killing the target organism [2]. Fipronil in foo mixtures is also known to cause serious diseases in the human body. Fipronil exposure changes blood biochemistry and sex hormone levels; it can also lead to reductions in the cellular immune response, antioxidant abilities, and carotenoid-reliant coloration [3].

Endocrine disruption occurs due to [.1] alterations in gene expression in Callinectes sapidus populations exposed to environmental levels of fipronil [4]. In 2017, according to a report by the European Commission's Health and Food Safety Directorate-General, levels of fipronil had been detected as high as 0.72 mg/kg in eggs and 0.77 mg/kg in chicken meat [5]. There are therefore many researchers interested in the residue analysis of fipronil in chicken eggs and muscles by liquid chromatography–tandem mass spectrometry (LC-MS/MS) [2,6,7] or fipronil detection using immunoassay [8,9] and gas chromatography [10]. Although the detection limit is extremely low, sample preparation is complex and time-consuming.

Recently spectroscopic detection of pesticide hazardous materials has paid much attention to the scientific community [11,12]. Raman methods have advantages of easy detection in aqueous solutions with the tremendous electromagnetic field enhancements on the metal surfaces to provide extreme high sensitivity to monitor organic contaminants [13,14]. Density functional theory (DFT) can be introduced to interpret Raman spectral features to understand the quantum mechanical properties of small organic molecules adsorbed on metal atoms [15].

SERS procedures for the detection of fipronil are unavailable in the literature[.2] , despite SERS being conventionally applied to biosensors [16-18] and the possibility of detecting trace amounts of toxic contaminants in minimally processed food products [19]. Due to significant enhancements and the resulting sensitivity, SERS has advantages over conventional methods (e.g., facile sample preparations and on-site detection). Because of irregular hot spots and equivocal selection rules, it is essential to identify novel nanostructures, such as oxide thicknesses on a Si/SiO2 [6] substrate or AgNPs [18,20,21], and convenient methods to increase sensitivity. The ability to increase both the sensitivity and selectivity of the SERS analytical method is the main reason for its role as a primary emergent spectral technique that has found many applications with food, such as monitoring and quality control in industrial food manufacture, food safety in agricultural plant production [22], and detection of pesticide residues in food [23].

DFT calculations have also seen significant advances in a wide range of molecular properties [24] to allow a close connection between theoretical and experimental research and often produces important evidence about the geometric shape, electronic detail, and spectroscopic properties of the issues being studied [25,26]. Therefore, the development and application of systems based on combinations of SERS and DFT calculations are a current hot topic that has contributed to better understanding of the phenomena related to detection and monitoring at very low concentrations—especially at physiological levels [17,21,23]. In particular, a great deal of effort has been reported toward developing a simple and rapid method for detecting the pesticide fipronil on eggshells and in liquid eggs by Raman spectroscopy [5]. However, some limitations associated with previously reported sensors remain, which need more facile assay techniques for fipronil detection to overcome them. Notably, there has been no recent significant consideration of SERS application combined with DFT calculations for fipronil detection.

The maximum fipronil level allowed in food (rice and corns) by the United States Food and Drug Administration and the European Commission is 0.01 ppm. Despite previous reports on the detection of fipronil using Raman spectroscopy or other methods, a more efficient nanostructure platform, in terms of facile usage and improved sensitivity, would be still helpful in analytical food and agriculture science. In this study, we report the detection of fipronil using SERS on AgNPs, considering both calculated vibrational wavenumbers and Raman intensity patterns. Our method can usefully detect fipronil in food, such as eggshells and liquid eggs, and in the water and soil environment.

2) Page 1 Lines 31-33: Authors previously suggested that “SERS procedures for the detection of fipronil are unavailable in the literature” however they then go on to detail and cite a case where SERS is employed to detect fipronil in eggshells and liquid eggs (Page 2 Lines 48-50). These statements are a contradiction and should be edited.

® Thank you for your comments. We are the first reported on fipronil detection using SERS on AgNPs. We referenced the article [5] below where the authors did not use SERS but employed normal Raman spectroscopy to detect fipronil in eggshells and liquid eggs as follows:

5. Tu, Q.; Hickey, M.E.; Yang, T.; Gao, S.; Zhang, Q.; Qu, Y.; Du, X.; Wang, J.; He, L. A simple and rapid method for detecting the pesticide fipronil on egg shells and in liquid eggs by Raman microscopy. Food Control 2019, 96, 16–21.

3) SERS based detection of pesticides are well reported in the literature and it is not fully clear what the novelty of the present manuscript is. While the merit of reporting specific details of fipronil detection with a SERS technique and analysis with DFT is certainly relevant, it is unclear whether the SERS protocol in the present manuscript has been developed by others in the literature or if it is new as suggested when the authors state “our method.” How does this SERS based protocol for fipronil detection compare and contrast to those already in the literature employing similar types of SERS detection? Specifically, has a similar SERS protocol been successfully employed with other pesticides, e.g. template-free AgNPs with CTAC? It is highly recommended the authors expand their introduction to include these items with comparisons to the literature of other pesticides.

® We are the first reported on fipronil detection using SERS on AgNPs and SERS procedures for the detection of fipronil are unavailable in the literature before. We could not compare our protocol for fipronil detection to those already in the literature employing similar types of SERS detection. We provided the novelty of the present manuscript as follows:

SERS signals of fipronil adsorbed on the surface of the AgNPs could not obtained, even when screening high concentrations (100–1,000 ppm) of pure fipronil [5]. This was mainly due to the weak interaction between fipronil and the AgNPs and the low solubility of fipronil, which can easily crystallize out of aqueous solutions. To overcome these limitations, a cationic surfactant, CTAC, was added to the mixture interaction solution of fipronil and AgNPs to improve dispersibility of fipronil in aqueous solutions. Figure 2a illustrates a plausible mechanism for the adsorption of fipronil molecules [.3] on AgNPs surfaces using CTAC linking. The surfaces of plasmonic AgNPs are modified by the CTAC molecules by electrostatic interactions between the positively charged CTAC and the negatively charged AgNPs. CTAC is a cationic surfactant that may be usable in the modification of the surface charge of AgNPs; it could also be used as a dispersant to prevent aggregation of AgNPs, while increasing the solubility of fipronil in solution. The binding between fipronil and AgNPs in the presence of CTAC is easier than in the absence of CTAC due to electrostatic interactions between the positively charged CTAC-modified AgNPs and the negatively charged fipronil. In general, fipronil molecules interacting with the surface of AgNPs through nitrogen atoms do so more strongly than other atoms in fipronil molecules. The atomic charges of these nitrogen atoms, such as N19, N20, and N24 (atomic numbering depicted in Figure 1a), were previously estimated at -0.58, -0.77, and -0.56 (a.u.), respectively, after fully unconstrained geometry optimization in aqueous phases [31]. Our DFT calculations of atomic charges of these nitrogen atoms were also consistent with that data. Thus, we can predict that the charge of fipronil is negative in the interaction with the AgNPs surface. Moreover, CTAC has a counter ion chloride in molecular leading to chloride-capped AgNPs allow for high-intensity SERS spectra of molecules to be obtained, it may also play a key role in understanding some fundamental principles behind SERS. Chloride was well known the chloride activation effect on the SERS spectra of some molecules such as phthalazine and pyridine adsorbed on silver colloids [33,34].

33. Muniz‐Miranda, M.; Sbrana, G. Evidence for surface Ag+ complex formation by an anion‐induced effect in the SER Spectra of phthalazine adsorbed on silver sols. J. Raman Spectrosc. 1996, 27, 105-110.

34. Otto, A.; Bruckbauer, A.; Chen, Y. X. On the chloride activation in SERS and single molecule SERS. J. Mol. Struct. 2003, 661, 501-514.

4) The authors quote the maximum residue limit of 0.01 ppm for fipronil in food set in the U.S. and Europe, however, values range depending on the type of tissue being sold as food. Can the authors clarify this point in the text?

® We clarified this point in the manuscript as follows:

“The maximum fipronil level allowed in food (rice and corns) by the United States Food and Drug Administration and the European Commission is 0.01 ppm.”

5) How does the Raman spectrum collected to compare to the literature reports of Raman detection of fipronil in or on eggs (ref 17)?

® Thank you for your comments. We changed from the reference number [17] to number [5] in the manuscript. We collected Raman spectrum to compare to the literature reports of Raman detection of fipronil in or on eggs [5] as follows:

Table S1. Comparison of this work with Raman microscopy method for detecting the pesticide fipronil on egg shells and in liquid eggs.

Nanostructure

Samples

Raman shift at   (cm-1)

Sensing method

LOD

Reference

Gold-coated glass slide

Liquid egg

2256

Raman

0.32 mg/kg

[5]

Egg shell

2256

Raman

0.065 mg/m2

AgNPs

Solution

2236

SERS

0.001 ppm

Present work

6) The authors suggest they attempted SERS experiments with AuNPs but were unable to observe any strong bands. The authors make no mention regarding the experimental materials or procedural details of creating their AuNPs nor do they report the size or the position of the UV-vis band. The authors must report these details of the AuNPs if a qualitative comparison is to be drawn. What are some possible reasons why AuNPs failed fipronil detection? Can these same problems occur with AgNPs leading to poor reliability in fipronil detection? Further justification of why AuNPs did not detect fipronil is required.

® We referred the article [5], the authors first considered using SERS which is known to be much more sensitive than Raman spectroscopy due to the enhancement produced by nano-roughened metallic surfaces. However, after several preliminary experiments using AuNPs and AgNPs, as well as cysteine-modified AuNPs, they could not achieve fipronil signals; even when screening high concentrations (100–1000 mg/L) of pure fipronil (with laser 780 nm). This was mainly due to the weak interaction between fipronil and the AuNPs and AgNPs, as well as the low solubility of fipronil which can easily crystallize-out of aqueous solutions. Based on significant differences in the solubility of fipronil in acetone (20°C, 545.90 g/L) and water (20°C, 1.90×10−3 g/L), they were inspired to develop a rapid and simple method to extract and recover fipronil crystals for detection using Raman microscopy. In our work, we prepared the AuNPs and AgNPs using a citrate reduction method, as well as CTAC-modified AuNPs and CTAC-modified AgNPs, also employed laser excitation 514 and 633 nm. But we could only obtain the SERS signal of fipronil with CTAC-modified AgNPs utilizing laser 514 nm. Maybe due to CTAC has a counter ion chloride in molecular and chloride-capped AgNPs allow for high-intensity SERS spectra of molecules to be obtained, it may also play a key role in understanding some fundamental principles behind SERS. Chloride was well known the chloride activation effect on the SERS spectra of some molecules such as phthalazine and pyridine adsorbed on silver colloids [33,34].

33. Muniz‐Miranda, M.; Sbrana, G. Evidence for surface Ag+ complex formation by an anion‐induced effect in the SER Spectra of phthalazine adsorbed on silver sols. J. Raman Spectrosc. 1996, 27, 105-110.

34. Otto, A.; Bruckbauer, A.; Chen, Y. X. On the chloride activation in SERS and single molecule SERS. J. Mol. Struct. 2003, 661, 501-514.

7) Page 5: Line 175-7: Authors state that there is “substantial adsorption of fipronil on the AgNPs surface. What evidence is there from the UV-vis spectra (Figure 2b) that fipronil is adsorbed onto the Ag nanoparticles and not just present in solution? Would a spectrum of CTAC-Fip (absent AgNPs) not also result in a band positioned at 280nm? The authors state there is an appearance of the 280 nm band in the AgNPs-CTAC-Fip spectrum suggesting fipronil is present in the mixture’s solution, but without further evidence (e.g. a shift in the 280nm band position) it is difficult to support the statement that fipronil is adsorbed on silver.

® We included the estimation of the loading efficiency and revised the manuscript as follows:

Figure 2b. UV–visible extinction spectra of initial fipronil (100 ppm), the supernatant solution of the excessive fipronil obtained by reaction with AgNPs-CTAC, subsequent centrifugation. and the removal of the supernatant fipronil from the precipitate AgNPs-CTAC-Fipronil. The extinction values were compared to yield the loading efficiency to estimate the amount of the adsorption of fipronil on AgNPs.

As illustrated in Figure 2b, the loading efficiency [35] of fipronil on AgNPs-CTAC was calculated using the UV-Vis absorption measurements from the initial free amount of fipronil subtracted by the supernatant solution of the excessive fipronil obtaining by incubation with AgNPs-CTAC then centrifugation and removal of precipitate AgNPs-CTAC-Fipronil on the basis of the following equation, where O.D. stands for optical density. The loading efficiency of fipronil on the AgNPs-CTAC was estimated to be ~38.09%.

8) UV-vis spectra should have the y-axis labeled as extinction. Extinction is the sum of both absorbance and scattering processes. The scattering component is often overlooked or ignored in the literature however the scattering process may still be significant here as observed in the UV-vis of the AgNPs (Figure 2b), therefore this plot would be more correctly termed extinction in the figure and the text.

® Thank you for your comments. We corrected the y-axis labeled as extinction in the manuscript as bellows (question 7).

9) Can the authors expand on the comment of testing for linearity, as it is an important parameter for determining the useful range of this technique? More generally, can the authors comment on the applicability of fitting the Raman intensity versus concentration to a linear line? Some other SERS based platforms for pesticide detection have reported linearity with double logarithmic scales (Freundlich adsorption of a solute on the surface of an adsorbent). The inset used by the authors to fit a linear line still appears to have a curvature as predicted in a Freundlich adsorption isotherm.

® We attempted to improve the fitting in Figure 4(b). At low concentrations of 0.0001–0.1 ppm, the linear fitting could be possible despite the plateau at high concentrations region above 50 ppm. Although it is not absolutely certain whether the adsorption behaviors should follow either Langmuir or Freundlich isotherms, out study may be helpful to estimate a trace of fibronil in aqueous solutions.

10) Can the authors include the value of the slope calculated in Figure 4B? Responsivity, or slope of a calibration curve, for a specific protocol, is a useful and therefore relevant parameter that should be reported in sensing.

® We provided the slope as newly added in Figure 4(b) of the revised manuscript.

Figure 4(b). The calibration curves of the vibrational band intensities at ~2236 cm−1, the inset shows fitting of the linear region of 0.0001–0.1 ppm. Each SERS experiment was repeated three times independence to make sure the reliability of this evaluation. The mean and standard deviation values were defined.

 11) The authors conclude in the main text that an energy difference of ~2 kcal/mol between binding at the nitrile and the N1 position to the Ag cluster is “negligible,” while in the abstract they suggest this same difference suggests binding through the nitrile is favorable, i.e. not negligible. Can the authors elaborate on these statements and the differences between the nitrile and N1 position? Further, how was binding between the fipronil and the Ag cluster set or optimized in the DFT calculations prior to calculation of the Raman spectra? The experimental details suggest the cluster was optimized separately from the free molecule, however, would interaction of the cluster and free molecule impact the optimized structures? Determining how fipronil is optimized with an Ag cluster will ultimately impact the resulting Raman spectrum calculated and therefore it is recommended the authors comment on this detail.

® We corrected in the manuscript as follows:

“AgNP modeling was conducted using the B3LYP/LANL2DZ method to optimize Ag6 cluster geometry prior to binding with the fipronil molecule, then the whole complex of Ag6-fipronil was fully optimized again.”

Figure S1. Simulated spectra of the adsorbed states of fipronil in the DFT calculations.

12) The bands in Figure 3a are difficult to distinguish due to their relative size compared to the nitrile band. Can an inset of the 400-1700 cm-1 regions be included? Can DFT predict a noticeable shift in the nitrile fipronil band upon the inclusion of the Ag6 cluster relative to the free molecule? Can the shift if the vibrational mode of the nitrile group be correlated with interaction with the AgNPs surface?

Figure 3(b). Simulated spectra of the adsorbed states of fipronil using the six cluster atoms at the DFT calculations.

 ® We added an inset of the 400-1700 cm-1 regions in Figure 3a in the manuscript as follows. DFT could predict a noticeable shift in the nitrile fipronil band upon the inclusion of the Ag6 cluster relative to the free molecule. Also, the shift if the vibrational mode of the nitrile group could be correlated with interaction with the AgNPs surface.

Specific Concerns

a. Is the experimental NR spectrum in Figure 1b baselined corrected? If so, this should be mentioned in the caption.

® We mentioned in the caption of Figure 1b in the manuscript as follows:

“Figure 1. ...(b) Experimentally measured and theoretically calculated Raman vibrations of fipronil, respectively. The baseline correction for normal Raman (NR) spectrum was recorded for the standard fipronil powder (laser 633 nm).”

b. Page 5 Line 152: Wrong units stated for Raman Shift “Unit in cm1” should be corrected to cm-1.

®Thank you for your comments. We changed from “Unit in cm1” to “Unit in cm-1” in the manuscript.

c. Figure 3b appears to show the molecular structure of the fipronil bound to the metal clusters through the nitrile moiety, in opposition to the stated structure in the caption as binding through the N1 atom. The authors must clarify.

®Thank you for your comments. We corrected the caption of Figure 3 in the manuscript as follows:

“Figure 3. (a) Theoretically estimated Raman vibrational spectra of fipronil binding at the nitrile moiety on Au6 and Ag6 atom clusters;...”,

 Finally, authors would like to appreciate the Reviewer #4 once again for giving valuable comments. Thank you.

Round 2

Reviewer 1 Report

I went through the reply of the authors, but I think, also in this second round of revision, that this paper is not suitable for publication in Sensors. The determination of a detection limit of a pesticide with silver nanoparticles is not enough for a publication (plenty of papers in literature do very similar things) unless it is integrated by an in-depth analysis of other aspects. In this case, the authors try to provide an explanation for the binding mechanism of fipronil to silver, but in my opinion their study is not sound enough to lead to reasonable conclusions.

1)    Additional DFT calculations have been carried out in Ag3+ and Ag4+ clusters, proposing different binding sites (N1 and CN). Looking at Figure S1, however, all spectra are significantly different from the experimental one and the attribution of the (Ag-fipronil binding CN) as the most likely situation is arbitrary: this spectrum is not clearly more similar to the experimental one with respect to the others.

3)    The whole DFT part is still in contrast with the model proposed in Figure 2a. The authors propose that fipronil is surrounded by positive charges and the interaction with the negatively charged silver surface is electrostatic. This is not compatible with a fipronil – silver surface bond occurring through the CN group: the CN group also bears a partial negative charge (figure 2b), reinforcing the idea that it is surrounded by the positive charges of the surfactant. The fact that CTAC has a weak Raman signal does not count: if it surrounds fipronil, it prevents the possibility of forming a bond through the CN group.

Author Response

Reply to Reviewer #1's Comments

The authors deeply appreciate the Reviewer #1 for giving helpful comments. We revised the manuscript to address the issues raised by the Reviewer #1 and corrected unclear expressions and errors in the manuscript. According to Reviewer #1’s comments, we revised the manuscript as follows:

I went through the reply of the authors, but I think, also in this second round of revision, that this paper is not suitable for publication in Sensors. The determination of a detection limit of a pesticide with silver nanoparticles is not enough for a publication (plenty of papers in literature do very similar things) unless it is integrated by an in-depth analysis of other aspects. In this case, the authors try to provide an explanation for the binding mechanism of fipronil to silver, but in my opinion, their study is not sound enough to lead to reasonable conclusions.

1)Additional DFT calculations have been carried out in Ag3+and Ag4+ clusters, proposing different binding sites (N1 and CN). Looking at Figure S1, however, all spectra are significantly different from the experimental one and the attribution of the (Ag-fipronil binding CN) as the most likely situation is arbitrary: this spectrum is not clearly more similar to the experimental one with respect to the others.

® According to the comments, we had attempted to calculate SERS spectra of fipronil adsorbed on Ag3+ and Ag4+ along with the Ag6 cluster. Compared to the NR spectrum, it is admitted that the SERS spectra could not perfectly match with our calculations. Multiple adsorption structures were presumably possible due to the small energy differences between the binding modes of fipronil on Ag, Our compiled calculation data suggested however that Ag6 cluster-fipronil (binding at CN) is the best similar from the experimental SERS.

3)The whole DFT part is still in contrast with the model proposed in Figure 2a. The authors propose that fipronil is surrounded by positive charges and the interaction with the negatively charged silver surface is electrostatic. This is not compatible with fipronil – silver surface bond occurring through the CN group: the CN group also bears a partial negative charge (figure 2b), reinforcing the idea that it is surrounded by the positive charges of the surfactant. The fact that CTAC has a weak Raman signal does not count: if it surrounds fipronil, it prevents the possibility of forming a bond through the CN group.

® We added a newly revised Figure 2a in the manuscript as follows:

Figure 2(a) Plausible adsorption mechanism of fipronil on AgNP surfaces;

English language and style are fine/minor spell check required

® We consulted the native speaker (#1533, UniLecture) to check our English in the revised manuscript as attached in the Author’s Notes to the Reviewer.

Finally, authors would like to appreciate the Reviewer #1 once again for giving valuable comments. Thank you.

Reviewer 4 Report

I believe the authors have made significant improvements to the present manuscript and most of the concerns raised in the previous review. I would recommend accepting for publication.

Author Response

The authors deeply appreciate the Reviewer #4 for giving positive comments. According to Reviewer #4’s comments, we revised the manuscript as follows:

I believe the authors have made significant improvements to the present manuscript and most of the concerns raised in the previous review. I would recommend accepting for publication.

->We appreciate the input that the Reviewer #4 has given, which definitely helped to improve our manuscript. The authors would like to thank Reviewer #4 for give valuable comments.

English language and style are fine/minor spell check required

->We consulted the native speaker (#1533, UniLecture) to check our English in the revised manuscript as attached in the Author’s Notes to the Reviewer.

Finally, authors would like to appreciate the Reviewer #4 once again for giving valuable comments. Thank you.
